# Accuracy, Bias, and Improvements in Mapping Crops and Cropland across the United States Using the USDA Cropland Data Layer

**Tyler J. Lark** [1,*] **, Ian H. Schelly** [1] **and Holly K. Gibbs** [1,2]

1  Nelson Institute Center for Sustainability and the Global Environment (SAGE),
   University of Wisconsin-Madison, Madison, WI 53726, USA; ianschelly@gmail.com (I.H.S.);
   hkgibbs@wisc.edu (H.K.G.)
2  Department of Geography, University of Wisconsin-Madison, Madison, WI 53726, USA
*  Correspondence: lark@wisc.edu

**Abstract:** The U.S. Department of Agriculture's (USDA) Cropland Data Layer (CDL) is a 30 m resolution crop-specific land cover map produced annually to assess crops and cropland area across the conterminous United States. Despite its prominent use and value for monitoring agricultural land use/land cover (LULC), there remains substantial uncertainty surrounding the CDLs' performance, particularly in applications measuring LULC at national scales, within aggregated classes, or changes across years. To fill this gap, we used state- and land cover class-specific accuracy statistics from the USDA from 2008 to 2016 to comprehensively characterize the performance of the CDL across space and time. We estimated nationwide area-weighted accuracies for the CDL for specific crops as well as for the aggregated classes of cropland and non-cropland. We also derived and reported new metrics of superclass accuracy and within-domain error rates, which help to quantify and differentiate the efficacy of mapping aggregated land use classes (e.g., cropland) among constituent subclasses (i.e., specific crops). We show that aggregate classes embody drastically higher accuracies, such that the CDL correctly identifies cropland from the user's perspective 97% of the time or greater for all years since nationwide coverage began in 2008. We also quantified the mapping biases of specific crops throughout time and used these data to generate independent bias-adjusted crop area estimates, which may complement other USDA survey- and census-based crop statistics. Our overall findings demonstrate that the CDLs provide highly accurate annual measures of crops and cropland areas, and when used appropriately, are an indispensable tool for monitoring changes to agricultural landscapes.

**Keywords:** accuracy assessment; accuracy metrics; map bias; confidence; crop maps; Cropland Data Layer; land use/land cover change; remote sensing products

## 1. Introduction

Mapping and monitoring crops and croplands can generate powerful insights about our environment and agricultural production systems [1–3]. Because satellite-based remote sensing products are able to efficiently capture land use/land cover (LULC) and their variations across space and time, these data are increasingly chosen as the basis for agricultural and environmental decision making, including policy creation, evaluation, and enforcement [4–8]. With the increased availability and use of detailed remotely sensed land cover products, however, there is a growing need to understand their accuracy and reliability for different applications [9–12].

In the United States, the Department of Agriculture's (USDA) Cropland Data Layer (CDL) is frequently utilized to monitor agricultural land due to its nationwide coverage, agricultural focus, and annual frequency [13–16]. Produced by the National Agricultural Statistics Service (NASS), this satellite-derived map has provided complete coverage of

the conterminous U.S. each year since 2008. Since it tracks specific crops at field-relevant resolutions, it is an ideal tool to detect geographic trends and changes in cultivation. Previous studies have used the CDL to track crop rotations and planting patterns [17–20], evaluate Farm Bill policies such as crop insurance and the Sodsaver program [8,21,22], and assess the environmental outcomes of various land management systems [20,23,24], among many other applications. Estimates of cropland area from the CDL are also used internally by NASS for a variety of reports and survey applications as well as considered by other government organizations such as the Environmental Protection Agency, for example, to monitor compliance with land protections in renewable energy policies [4,25].

To characterize the CDL's performance, NASS calculates land cover class-specific accuracies at the state level and releases them with each annual state CDL product [26]. These estimates are based on a comparison with parcel level data from the USDA Farm Service Agency (FSA) [27] and another land cover map, the National Land Cover Dataset (NLCD) [28,29]. While these comparisons provide insights into the accuracy of the CDL for a given state and year, applications of the CDL product typically extend well beyond this scope; many analyses utilize modifications of the original CDL datasets, compare across the state products, and/or estimate changes in LULC over time [30–34]. Despite the prevalence of these applications, the performance of the CDLs in many of these extensions has not been evaluated.

Given this lack of evaluation, several articles have questioned the reliability of analyses that use CDL data to identify recent agricultural trends, citing concerns about both the CDL's accuracy and its appropriateness for measuring changes to the landscape [35–40]. Such critiques often cite low reported accuracies for the CDLs when mapping certain crops in specific regions or when depicting nonagricultural land covers such as grasslands. Despite the potential validity of these concerns, all such critiques to date have lacked a systematic nationwide assessment of the CDL accuracy beyond comparisons with coarse data, thereby leaving substantial uncertainty surrounding the CDL's ultimate dependability. Furthermore, select approaches for measuring LULC change using the CDL and other land cover products may help overcome some of the CDL's limitations and improve analysis outcomes [41], though the efficacy of these techniques has not yet been fully quantified. For example, aggregating specific land cover classes into broader domains, such as cropland and non-cropland, can help address low classifier accuracies of specific cover classes by eliminating errors associated with distinguishing different crop types and among various non-cropland covers, such as the many grassland categories historically delineated in the CDL [30,40,42].

In this paper, we comprehensively quantified the accuracy of the CDL at the national scale and evaluated the outcomes relevant for applications of the CDL for mapping crops and cropland. First, we investigated the benefits of consolidating classes within remote sensing products and quantified the CDL's ability to distinguish between crop and non-cropland covers at multiple spatial scales and thematic resolutions. Then, we calculated nationwide accuracies for both specific and aggregate classes of the CDL and mapped the spatial variation in accuracies across the U.S. based on congruence with FSA and NLCD data. We then explored the use of pixel-level classifier confidence information to provide additional higher-resolution understanding of thematic certainty. Finally, we estimated the annual bias in mapping specific crops within the CDL and derived new, bias-adjusted area estimates for the major crop types. We conclude with a discussion of the implications of these analyses with a particular focus on recommendations for improving LULC change analyses.

## 2. Materials and Methods

### 2.1. Overview of Assessed and Reference Datasets

The Cropland Data Layer is a crop-specific land cover map produced annually by the USDA National Agricultural Statistics Service (NASS). Complete coverage of the conterminous United States dates back to 2008, while some states and years predate the

nationwide product. Primary satellite imagery inputs for the CDL vary according to availability and effectiveness but have included the Resourcesat-1 Advanced Wide Field Sensor (AWiFS), Resourcesat-2 Linear Imaging Self Scanning (LISS), Landsat-5 Thematic Mapper (TM), Landsat-7 Enhanced TM Plus (ETM+), Landsat-8 Optical Land Imager (OLI), Sentinel-2 A/B, and Deimos-1 and UK-2 from the Disaster Monitoring Constellation. Input images are collected and used internally by NASS throughout the growing season, and the final, publicly released CDL is intended to capture the area and geospatial distribution of crops in midsummer. Data processing and classification generally occur independently at the state level by NASS analysts, and the nationwide CDL mosaic that results contains up to 155 classes of cultivated crops and 23 classes of non-cropland covers. Most states, however, contain a smaller subset of applicable classes, typically fewer than 30 crops and a dozen non-crop covers [26].

In producing the CDL, NASS uses supplementary information from both the FSA and the USGS. Specifically, NASS leverages a selection of data from the FSA's Common Land Unit (CLU) administrative database to train all cultivated crop classes of the CDL and assess their accuracy. CLU data are collected and confirmed by USDA County Field Service Centers and constitute a comprehensive geospatially tagged database of all land owned by agricultural producers who participate in an FSA program [27]. This represents the most complete dataset on U.S. agricultural land use, but is not available to the public [16].

For training and assessing non-cropland cover categories, NASS uses the USGS-led NLCD as a reference [26,43]. The NLCD is a nationwide 30-meter resolution, 20-class land cover map that follows a modified Anderson level I/II classification system [44]. The product's mapping emphasizes non-cropped vegetative areas, and was historically produced for 5-year epochs, though the most recent product release has improved coverage to 2–3-year intervals. It should be noted that while the NLCD is used as an input in training the CDL classifier, the CDL does not simply revert to the NLCD in non-crop locations. Instead, the CDL incorporates the NLCD and other data to generate its own unique mapping of non-crop areas.

During the assessment of the CDL, NASS produces and publishes online the confusion matrices used to determine the reported accuracies. Referred to as the "error supermatrices," these datasets are generated each year at the state or multistate level and report the number of times specific CDL classes were mapped either consistently or inconsistently against CLU data from the FSA for all cultivated crops, or against the NLCD for non-cultivated land covers [26,27]. While the FSA data and NLCD provide valuable references for comparison, each differs from traditional reference data used for land cover map evaluation. In particular, the FSA data are not selected via a probability sampling design. In addition, because the dataset is generated for other USDA programmatic purposes, its classes do not always align perfectly with the classes of the CDL, leading to potential mismatch between the target and reference data. Nevertheless, the FSA dataset represents an incredibly rich and extensive source of reference information that is of a quality rarely available for remote sensing accuracy assessments. The NLCD, as a satellite-based land cover map, is not fully independent nor necessarily more accurate than the CDL. The NLCD is also not produced annually, such that the closest NLCD product available at the time of CDL production must be utilized, leading to potential temporal mismatch between the target and reference data. Despite these limitations, these two datasets provide powerful points of comparison for understanding how CDL performance varies across space and time.

### 2.2. Investigating Effects of Aggregation: Superclass and Consolidated Class Accuracies

We used the data reported in the CDL error supermatrices to derive supplemental accuracy metrics useful for characterizing and understanding the CDL across scales and applications. A summary and example of each accuracy metric we assessed is presented in Table 1, with further details of their derivation described in the section below.

**Table 1.** Accuracy metrics, measured classes, and associated examples. The table describes each of the four main metrics reported in this paper and provides an example of each metric from the producer's accuracy and user's accuracy perspectives.

| Metric: | Reported For: | Measures Accuracy of Identifying: | Producer's Example | User's Example |
|---|---|---|---|---|
| **Class Accuracy** | Specific classes | Specific classes | The likelihood that actual corn is mapped as corn | The likelihood an area mapped as corn is actually corn |
| **Superclass Accuracy** | Specific classes | An aggregated domain | The likelihood that actual corn is mapped as cropland | The likelihood an area mapped as corn is actually cropland |
| **Consolidated Class Accuracy** | An aggregated domain | An aggregated domain | The likelihood that actual cropland is mapped as cropland | The likelihood an area mapped as cropland is actually cropland |
| **Average Class Accuracy** | An aggregated domain | Specific classes | The likelihood that any crop is mapped as that specific crop | The likelihood that any mapped crop is actually that crop |

Initially, NASS treats their reference data as a simple random sample and calculates the class accuracies for all specific land cover classes within each state according to the general formula:

$$Class\ Accuracy_x = \frac{Pixels\ correct_x}{Pixels\ total_x} \qquad (1)$$

for each specific crop *x*, where *pixels correct* is the number of mapped pixels that match the reference data in a given region, and *pixels total* is either the total number of reference data observations (for calculating producer's accuracy) or mapped pixels (for calculating user's accuracy) for each class. Producer's accuracies reflect errors of omission; they indicate how likely a feature is to be correctly captured by the remote sensing product. User's accuracies reflect errors of commission, and indicate how likely a mapped class correctly resembles features on the landscape [45].

Aggregating land cover classes to broader thematic classes increases accuracy by lowering thematic specificity [28,46]. To understand how well the CDL can distinguish general cropland from non-cropland areas, we assessed the accuracy of aggregated cropland and non-cropland domains as delineated in Lark et al. (2015), based on original NASS distinctions [16,26]. The aggregated cropland category includes all annually cultivated row, closely planted, and horticultural crops as well as tree crops and actively tilled fallow (Appendix A Table A1). The non-cropland domain includes all remaining CDL classes.

First, we calculated how frequently each *specific* class of the CDL is mapped as any class within the correct cropland or non-cropland domain. We refer to this as the superclass accuracy for each specific class, and derived it as

$$Superclass\ Accuracy_{C,x} = \frac{Pixels\ in\ correct\ domain_C}{Pixels\ assessed_x} \qquad (2)$$

for each specific class *x* included in the domain *C* (e.g., cropland or non-cropland). For the cropland domain, the superclass producer's accuracy indicates how frequently a specific crop on the landscape (e.g., corn) was mapped by the CDL as any type of crop in the cropland domain. The corresponding superclass user's accuracy represents how likely a pixel mapped as a specific crop was actually any type of crop (i.e., cropland) on the landscape.

From the relationship between specific class accuracy and superclass accuracy, it is possible to quantify the relative number of mapping errors where confusion occurs with another class within the same broader domain. We define this metric, which we refer to as the within-domain error rate, as the difference between a class's error rate and its

superclass error rate, normalized by the class error rate. It can also be derived directly from the previously calculated accuracy metrics as

$$Within\ Domain\ Error\ Rate_{C,x} = \frac{Superclass\ Accuracy_{C,x} - Class\ Accuracy_x}{1 - Class\ Accuracy_x} \quad (3)$$

for each specific class *x* included in the domain *C*.

Then, we calculated the overall consolidated class accuracy for the entire cropland domain according the following equation:

$$Consolidated\ Class\ Accuracy_C = \frac{\sum_{x \epsilon C}(area_x \times Superclass\ Accuracy_{C,x})}{\sum_{x \epsilon C}(area_x)} \quad (4)$$

where *x* is each specific class belonging to the set of all classes in domain C, *area* is the area of class *x*, and *superclass accuracy* is the value calculated in Equation (2) above. Because the superclass accuracies give the likelihood that a specific class will correctly identify the broader domain, taking the area-weighted mean of the superclass accuracies across all classes within a domain depicts the likelihood that *any* class in a domain will correctly identify the broader domain. For the consolidated cropland domain, this calculation generates a single value that represents the accuracy with which the CDL can identify cropland in a given state and year. The user's accuracy for consolidated cropland represents the likelihood that any randomly selected pixel mapped as cropland in the CDL is actually cropland on the landscape. The producer's accuracy for consolidated cropland is the likelihood that cropland on the landscape is correctly mapped as cropland in the CDL. In similar fashions, Equations (2) and (4) can be used to calculate superclass accuracies for each specific non-crop class and for the single consolidated non-cropland domain.

For thoroughness and comparison, we also calculated the average specific class accuracy for each domain, according to the following equation:

$$Average\ Class\ Accuracy_C = \frac{\sum_{x \epsilon C}(area_x \times Class\ Accuracy_{C,x})}{\sum_{x \epsilon C}(area_x)} \quad (5)$$

The average specific class accuracy indicates how accurately, on average across the full domain, a randomly selected class is mapped in a given year. Tracking the average specific class accuracy across several years can thus indicate how well the CDL historically performed and improved over time at delineating specific crops.

### 2.3. Calculating Nationwide Accuracies

We next estimated nationwide accuracies for each original CDL class as well as for the newly derived aggregated metrics. To calculate nationwide accuracies, we weighed each state accuracy to account for disproportionate class areas and reference observations. For specific class accuracies of the original CDL, we normalized according to the following equation:

$$Nationwide\ Accuracy_x = \frac{\sum_{i \epsilon S}(Accuracy_{x,i} \times area_{x,i})}{\sum_{i \epsilon S}(area_{x,i})} \quad (6)$$

where *S* is the set of states or multistate regions for which data are produced in a given year, *Area* is the total area of class *x* mapped within the state or region *i*, and *accuracy* is the user's or producer's accuracy (Equation (1)) for region *i*. Similarly, Equation (6) was used to calculate the nationwide superclass accuracies for each crop by replacing the specific class accuracies with the appropriate superclass accuracies derived from Equation (2) above.

To derive the nationwide accuracies for consolidated land cover classes, we also area-weighed by each constituent class. This accounted for unequal areas of each class within the consolidated domain and ensured proportional contributions to the accuracy of the combined class. We considered only classes for which accuracy data existed when

summing class accuracies and areas, since failure to exclude the area of classes without data would falsely skew the nationwide mean values. Using the available data, we calculated nationwide accuracies for consolidated classes using the following formula:

$$Nationwide\ Consolidated\ Accuracy_C = \frac{\sum_{x \in C}(Nationwide\ Superclass\ Accuracy_x \times nationwide\ area_x)}{\sum_{x \in C}(nationwide\ area_x)} \tag{7}$$

Using the specific class accuracy in this formula gives the nationwide-specific class accuracy averaged across all land covers in the broader domain. Specifically:

$$Nationwide\ Average\ Class\ Accuracy_C = \frac{\sum_{x \in C}(Nationwide\ Accuracy_x \times nationwide\ area_x)}{\sum_{x \in C}(nationwide\ area_x)} \tag{8}$$

### 2.4. Mapping Spatial Patterns of CDL Accuracy and Confidence

We mapped a composite of all state- and class-level users' and producers' accuracies for each specific crop and non-cropland cover to better understand how the accuracy of CDL data varies spatially across the U.S. To generate these maps, each original CDL pixel was assigned the value of its specific class accuracy for that state and year and rounded to the nearest integer to facilitate storage as an eight-bit raster. We also mapped and delineated crop and non-crop components of the CDL confidence layer, which was provided courtesy of USDA NASS. The confidence layer is a coproduct of the remote sensing classification process and provides a measure of how well a specific pixel fits within the decision tree ruleset used to classify it [26,47]. A unique benefit of the confidence dataset is that it provides an independent value for each individual pixel, rather than a single value for all pixels of a given class within a state. It thus varies at the pixel level, enabling improved spatial understanding of expected errors within the CDL product [26].

We then combined the assessed accuracy and classifier confidence data into a single metric of CDL certainty to better understand the spatial variation in CDL performance. By integrating the pixel-resolution confidence layer into the state- and class-resolution accuracy estimates, a combined metric may offer additional insight or improved spatial representation of expected errors compared to standalone accuracy indicators. This is similar to the approach of using posterior probability spaces in change vector analysis [48].

We considered several ways to combine the accuracy and confidence data, including multiplying the two components (Equation (9)), averaging them (Equation (10)), and additional more elaborate combinations (e.g., Equation (11)):

$$Certainty = Class\ Accuracy \times Pixel\ Confidence \tag{9}$$

$$Certainty = \frac{(Class\ Accuracy + Pixel\ Confidence)}{2} \tag{10}$$

$$Certainty = Class\ Accuracy \left[1 + \frac{(Pixel\ Confidence) - (Average\ Class\ Confidence)}{(Average\ Class\ Confidence)}\right] \tag{11}$$

The approaches of Equations (9) and (10) benefit from their simplicity and intuitiveness. In Equation (11), the confidence data are used as a scalar multiplier to modify the class-level accuracy: if a pixel is mapped more confidently than the average of the other pixels in its class, then its certainty value will be greater than its class accuracy; if a pixel is mapped less confidently than average, then its certainty value will be lower than its class accuracy. Ultimately, the selection of a formula should be based on the needs of the specific application [48]. Thus, we present results only from the simple product combination (Equation (9)) in order to illustrate the concept and potential value of combining accuracy and confidence data but leave further investigation to future work and specific applications.

### 2.5. Estimating Map Biases and Bias-Adjusted Crop Acreages

Due to misclassifications within remote sensing products, area estimates derived directly from pixel counts are likely to be incorrect and either over- or under-predict actual

class area. Using data derived from confusion matrices, it is possible to quantify this bias relative to the reference data and subsequently make bias-adjusted area estimates accordingly [10,49]. While best practices in accuracy assessment stipulate the use of bias adjusted estimators with a probability sampling design [12,50], a simplified estimate of map bias and adjusted area estimates may still be derived and useful for products such as the CDL, where a large and high quality—though non-probabilistic—reference dataset is available. To illustrate this, we calculated the nationwide relative bias of each crop using the producer's and user's accuracy:

$$Simple\ Bias_x = \frac{Producer's\ Accuracy_x}{User's\ Accuracy_x} - 1 \tag{12}$$

for each class $x$ where the producer's and user's accuracies were those derived in Eq 6. This indicator of bias is equivalent to the number of assessed pixels mapped as class $x$ divided by the number of assessed pixels classified as class $x$ in the reference data, such that it reflects the relative over- or under-mapping of a class compared to the reference data. We then calculated bias-adjusted area estimates for each class $x$ by scaling the raw CDL acreage estimates by the amount of over or underprediction suggested by the bias:

$$Bias\ Adjusted\ Area_x = Class\ Area_x - (Class\ Area_x \times Simple\ Bias_x) \tag{13}$$

where *Class Area* is the area estimate for each class x derived from pixel counting and the *Simple Bias* is that derived in Equation (12).

## 3. Results

We first present results from our nationwide analysis of specific class accuracies, followed by nationwide results for the aggregated superclass and consolidated class metrics. Throughout the results section, we focus on data for the year 2012 as an example because it represents an intermediate year within the CDL's modern era of nationwide coverage, it was used in multiple applications [22,35,51], and it aligns well with the Census of Agriculture, the Natural Resources Inventory, and other intermittent data sources often used for comparisons with the CDL. The year 2012 was also particularly challenging for mapping agricultural LULC—moderate resolution imagery was limited, and a severe drought impacted crop development in many regions—such that our findings should be considered a conservative estimate of the performance of the CDL. For completeness, results were also generated for all years of nationwide CDL coverage 2008–2016 and have been reposited online as companion datasets at https://doi.org/10.5281/zenodo.4579863 (accessed on 1 January 2021).

### 3.1. Nationwide Accuracy of Specific CDL Classes

Nationwide area-weighted accuracies for the major crop classes of the CDL are generally very high. In 2012, corn, soybeans, and winter wheat—the three largest crops by area—were mapped correctly 95, 94, and 92% of the time from both the producer's and user's perspectives. The top 20 CDL land cover classes by area and their associated producer and user accuracies for 2012 are presented in Table 2, with accuracies for all 130 assessed land cover classes for 2012 included in Appendix A Table A2.

Overall, 10 crops had nationwide producer's accuracies of 90% or greater in 2012. These included sugarcane (97%); rice (96%); corn (95%); soybeans (94%); sugarbeets (94%); canola (94%); winter wheat (92%); cotton (91%); almonds (91%); and cranberries (91%). Five additional crops had class producer's accuracies higher than the average for all crops, 88.7%, and the remaining 90 crops with computable accuracies fell below the average class accuracy. In the same year, 17 crops had nationwide user's accuracies of 90% or greater (Appendix A Table A2). The remaining 88 crops had user's accuracies below the average of 90.3%. The disproportionate number of crops with below-average accuracy reinforces observations that the CDL performs best for major crops (defined by area) and less so for

minor crops. To this end, the 10 crops with the highest producer's accuracies made up 71.5% of the total mapped crop area.

**Table 2.** Nationwide class accuracies of major individual land covers in the 2012 Cropland Data Layer (CDL). Table shows area-weighted national average accuracies for the 20 most common classes by area in the 2012 CDL, calculated according to Equation (6), based on data from USDA National Agricultural Statistics Service (NASS). National accuracies of all crops and land covers for 2012 are listed in Appendix A Table A2.

| Class Name | ID | Producer Accuracy | Omission Error | User Accuracy | Commission Error |
|---|---|---|---|---|---|
| Corn | 1 | 95% | 5% | 95% | 5% |
| Cotton | 2 | 91% | 9% | 89% | 11% |
| Soybeans | 5 | 94% | 6% | 94% | 6% |
| Spring Wheat | 23 | 89% | 11% | 87% | 13% |
| Winter Wheat | 24 | 92% | 8% | 92% | 8% |
| Alfalfa | 36 | 75% | 25% | 80% | 20% |
| Other Hay/No Alfalfa | 37 | 57% | 43% | 57% | 43% |
| Fallow/Idle Cropland | 61 | 69% | 31% | 79% | 21% |
| Open Water | 111 | 90% | 10% | 81% | 19% |
| Developed/Open Space | 121 | 89% | 11% | 61% | 39% |
| Developed/Low Intensity | 122 | 83% | 17% | 74% | 26% |
| Developed/Med Intensity | 123 | 84% | 16% | 81% | 19% |
| Barren | 131 | 74% | 26% | 75% | 25% |
| Deciduous Forest | 141 | 88% | 12% | 75% | 25% |
| Evergreen Forest | 142 | 87% | 13% | 73% | 27% |
| Mixed Forest | 143 | 44% | 56% | 51% | 49% |
| Shrubland | 152 | 87% | 13% | 71% | 29% |
| Grassland/Pasture | 176 | 79% | 21% | 50% | 50% |
| Woody Wetlands | 190 | 70% | 30% | 63% | 37% |
| Herbaceous Wetlands | 195 | 61% | 39% | 47% | 53% |
| Average of All Crops | N/A | 88.7% | 11.3% | 90.3% | 9.7% |
| Average of All Non-Crops | N/A | 82.4 % | 17.6% | 69.4% | 30.6% |

Reported accuracies of specific non-crop classes of the CDL were generally lower than those of major crops (Table 2). However, it is important to acknowledge that the reported figures do not represent congruence with a verified ground or truth dataset of non-cropped areas, but rather are assessed against a reference dataset consisting of both FSA administrative crop data and the NLCD, itself a remotely sensed land cover map subject to misclassifications. Nonetheless, the lower levels of reported accuracy in the CDL non-crop classes suggest higher levels of uncertainty and potential error in the product and/or reference data, particularly when compared to the high-performance crop classes. The specific categories of open-, low-, and medium-intensity developed land as well as deciduous and coniferous forest, shrubland, and open water were all mapped with nationwide accuracies of greater than 80 percent, whereas specific classes of herbaceous and woody wetlands and grassland/pasture had lower nationwide performance that ranged from 47–79% (Table 2).

### 3.2. Consolidated Cropland and Non-Cropland Accuracies

Specific land cover classes of the CDLs are often combined into aggregated categories for applications such as measuring cropland area or conversion between major land cover types. As an example of aggregation, we assessed the accuracy of consolidated cropland and non-cropland domains across the U.S. from 2008–2016.

The area- and class-weighted nationwide accuracies for consolidated cropland in 2012 were 95.0% (producer's) and 97.4% (user's). Accuracies for the consolidated non-cropland domain were 97.8 and 88.8%, respectively. Consolidated classes also performed consistently well across time (Table 3). For example, in 2008—the oldest year for which nationwide data were produced—cropland user and producer accuracies were 95% and 98%, respectively.

**Table 3.** Average specific class and consolidated class accuracies for each year of the CDL. Data from USDA NASS (2016) based on the comparison of CDL with data from Farm Service Agency (FSA) and National Land Cover Dataset (NLCD) and processed according to equations (7) and (8). Cropland and non-cropland domains based on class distinctions in Appendix A Table A1.

| Metric | Type | 2008 | 2009 | 2010 | 2011 | 2012 | 2013 | 2014 | 2015 | 2016 |
|---|---|---|---|---|---|---|---|---|---|---|
| Average Crop Accuracy | Prod: | 88% | 89% | 89% | 89% | 89% | 89% | 90% | 90% | 92% |
| | User: | 90% | 90% | 91% | 91% | 90% | 91% | 92% | 91% | 92% |
| Average Non-Crop Accuracy | Prod: | 82% | 82% | 81% | 82% | 82% | 82% | 81% | 85% | 85% |
| | User: | 63% | 64% | 65% | 61% | 69% | 67% | 69% | 82% | 82% |
| Consolidated Cropland Accuracy | Prod: | 95% | 95% | 95% | 95% | 95% | 96% | 96% | 96% | 98% |
| | User: | 98% | 98% | 97% | 98% | 97% | 99% | 99% | 98% | 99% |
| Consolidated Non-Cropland Accuracy | Prod: | 97% | 97% | 98% | 98% | 98% | 98% | 97% | 99% | 99% |
| | User: | 84% | 85% | 89% | 82% | 89% | 87% | 89% | 98% | 98% |

In 2012, 30 of the 40 state or multistate assessment regions of the CDL had consolidated cropland producer's accuracies of 90% or greater (Appendix A Table A3). On the user's sides, all but two states—New York and Pennsylvania—mapped cropland correctly 90% of the time or greater. Oklahoma (OK) and Arizona (AZ)—more arid states where cropland contrasts with the surrounding landscape and is often irrigated—had the highest cropland user's accuracies, with values over 99%. More broadly, states with greater amounts of cropland typically had higher consolidated cropland accuracies (Figure 1), though this effect appeared to saturate beyond a certain threshold of crop area (e.g., 5 million acres). Similar trends were also observed when assessed by proportion (rather than total area) of cropland within each state [16,35].

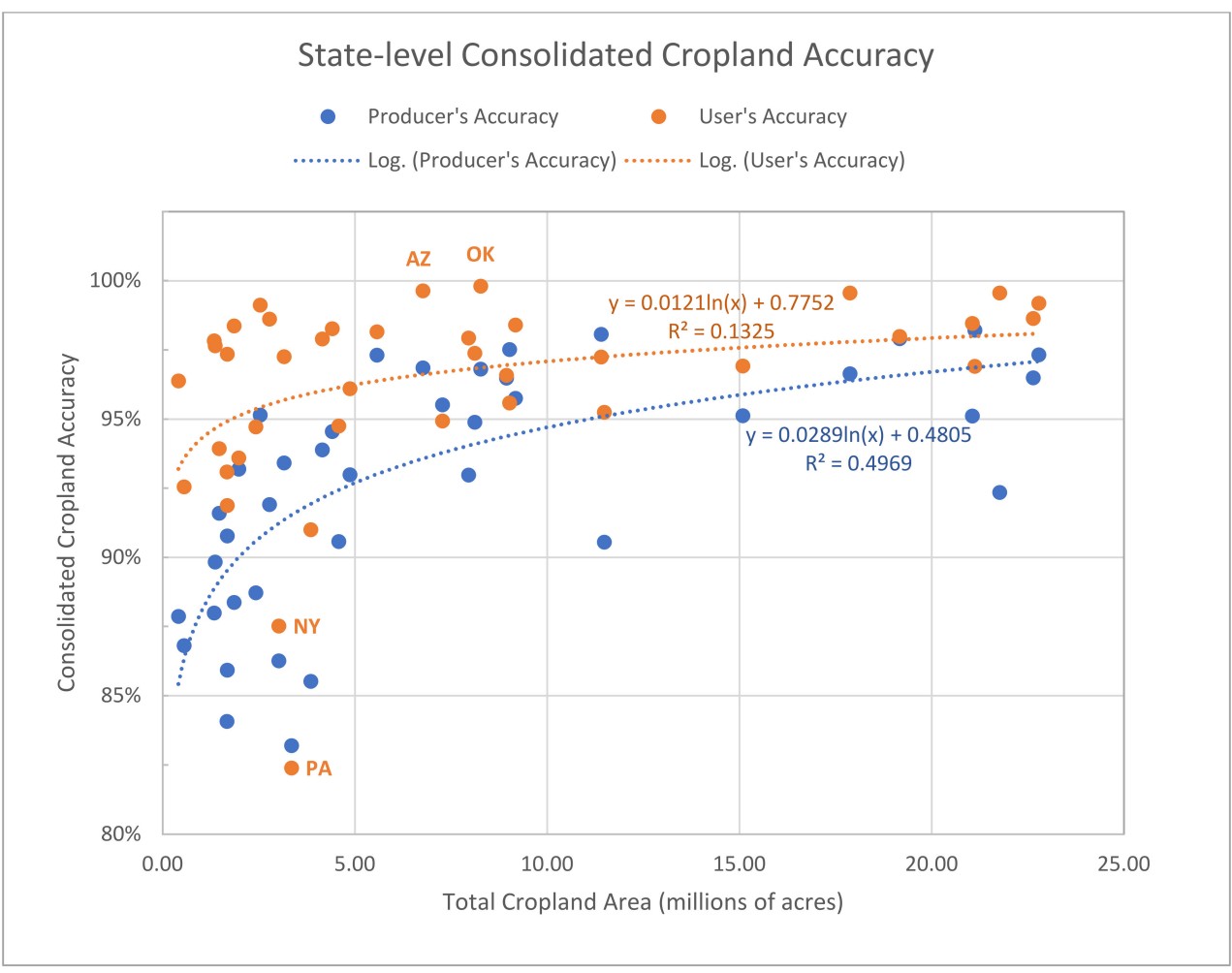

**Figure 1.** Plot of consolidated cropland user and producer accuracies for each state for 2012. Accuracies plotted against total crop area in each state. States with greater amounts of cropland typically had higher consolidated cropland accuracies. Plotting accuracy against the proportion of cropland within each state generated similar trends (data not shown).

### 3.3. Superclass Accuracies of Specific Crops and Land Covers

Within the aggregated domains, certain classes are more (or less) likely to align with their broader domain. Among crops mapped in the CDL with greater than one million acres, rice was the most accurate predictor of cropland on the landscape and most likely to be correctly identified as cropland, having superclass user's and producer's accuracies both over 99% in 2012 (Table 4). Areas of corn, the most prevalent crop, were labeled as cropland by the CDL 98% of the time in 2012 (superclass producer's accuracy), and pixels mapped as corn in the CDL were actually cropland on the landscape 98.5% of the time (superclass user's). Fields of alfalfa, oats, and fallow/idle cropland, on the other hand, were correctly labeled as cropland by the CDL just over 80% of the time. On the user's side, alfalfa was the only low outlier, yet still had an 86% superclass user's accuracy for the cropland domain.

Within the non-cropland domain, most superclass accuracies were high, with only a few exceptions (Table 5). Developed/Open Space was incorrectly mapped in locations that were actually cropland 25% of the time in 2012. Grassland/Pasture had an even lower user's accuracy and was mapped in cropped locations 32% of the time that year. Furthermore, the high ratio of superclass producer's accuracy to superclass user's accuracy—indicative of bias—in each of these classes suggests they are both considerably overmapped in locations that are actually cropland.

**Table 4.** Superclass producer and user accuracies for the top 20 classes by area in the cropland domain in 2012 as well as the relative rate of within-domain errors. Superclass accuracy is the likelihood that a given crop is identified correctly as cropland. Percentage of errors within domain is the proportion of errors in the original CDL where the confusion occurs among two crops within the cropland domain, rather than between a crop and non-cropland cover.

| CDL ID | Crop Class | CDL Acreage | Superclass Accuracy | | % of Errors within Domain | |
|--------|-----------|-------------|------------|---------|-------------------|---------------------|
| | | | Producer's | User's | Omission Errors | Commission Errors |
| 1 | Corn | 94,983,301 | 98% | 99% | 57% | 73% |
| 2 | Cotton | 13,114,321 | 98% | 100% | 77% | 96% |
| 3 | Rice | 2,671,894 | 99% | 100% | 84% | 95% |
| 4 | Sorghum | 6,262,444 | 96% | 99% | 81% | 95% |
| 5 | Soybeans | 69,810,086 | 98% | 99% | 65% | 76% |
| 6 | Sunflower | 1,595,069 | 94% | 99% | 52% | 79% |
| 10 | Peanuts | 1,657,438 | 98% | 99% | 88% | 93% |
| 21 | Barley | 2,852,300 | 94% | 99% | 78% | 92% |
| 22 | Durum Wheat | 1,860,552 | 98% | 99% | 92% | 97% |
| 23 | Spring Wheat | 12,303,171 | 96% | 99% | 64% | 92% |
| 24 | Winter Wheat | 34,784,199 | 97% | 99% | 55% | 88% |
| 26 | Dbl Win-Wht/Soybeans | 5,311,121 | 97% | 98% | 73% | 87% |
| 28 | Oats | 1,285,192 | 81% | 93% | 67% | 84% |
| 31 | Canola | 1,700,926 | 97% | 99% | 53% | 86% |
| 36 | Alfalfa | 16,167,152 | 80% | 86% | 27% | 40% |
| 41 | Sugarbeets | 1,238,159 | 99% | 100% | 77% | 94% |
| 42 | Dry Beans | 1,743,309 | 97% | 99% | 84% | 94% |
| 61 | Fallow/Idle Cropland | 24,395,076 | 80% | 92% | 35% | 67% |
| 69 | Grapes | 1,136,718 | 96% | 98% | 69% | 82% |
| 75 | Almonds | 1,155,344 | 98% | 99% | 78% | 94% |

Overall, the high superclass accuracies of non-crop classes compared to their low specific class accuracies reported in Table 2 suggests that a sizable portion of the mapping errors result from within-domain confusion among the various non-crop classes, rather than between non-cropland covers and crops. To quantify this, we calculated the relative within-domain error rate for each CDL class. This metric indicates what percentage of mapping errors were a result of confusion within the same domain. For example, corn had a relative within-domain omission error rate of 57% in 2012, which means that slightly more than half of the missed (i.e., omitted) corn fields were mapped as another crop in the CDL, rather than mapped as a non-cropland cover (Table 4). The within-domain proportion of commission errors for corn was 73%, which indicates that roughly three-quarters of all pixels that were incorrectly mapped as corn in the CDL were actually another crop on the landscape rather than a non-cropland cover.

**Table 5.** Superclass producer and user accuracies for all 16 classes in the non-cropland domain in 2012. Superclass accuracy is the likelihood a given class is correctly identified as non-cropland. Percentage of errors within domain is the proportion of errors in the original CDL where the confusion occurs among two land covers within the non-cropland domain, rather than between a crop and non-cropland cover.

| CDL ID | Land Cover Class | CDL Acreage | Superclass Accuracy | | % of Errors within Domain | |
|---|---|---|---|---|---|---|
| | | | Producer's | User's | Omission Errors | Commission Errors |
| 37 | Other Hay/Non Alfalfa | 23,881,755 | 89% | 86% | 68% | 62% |
| 92 | Aquaculture | 203,750 | 87% | 84% | 58% | 17% |
| 111 | Open Water | 32,373,788 | 99% | 95% | 89% | 76% |
| 112 | Perennial Ice/Snow | 427,601 | 100% | 99% | 100% | 97% |
| 121 | Developed/Open Space | 64,041,431 | 97% | 75% | 72% | 41% |
| 122 | Developed/Low Intensity | 28,380,971 | 99% | 91% | 96% | 69% |
| 123 | Developed/Med Intensity | 11,279,299 | 100% | 96% | 98% | 81% |
| 124 | Developed/High Intensity | 3,900,690 | 100% | 98% | 99% | 87% |
| 131 | Barren | 20,800,191 | 99% | 96% | 96% | 87% |
| 141 | Deciduous Forest | 239,843,277 | 100% | 97% | 94% | 89% |
| 142 | Evergreen Forest | 249,399,532 | 100% | 99% | 99% | 96% |
| 143 | Mixed Forest | 29,952,005 | 100% | 99% | 100% | 98% |
| 152 | Shrubland | 429,532,225 | 99% | 89% | 89% | 64% |
| 176 | Grassland/Pasture | 383,816,367 | 93% | 68% | 66% | 37% |
| 190 | Woody Wetlands | 75,447,681 | 99% | 93% | 97% | 83% |
| 195 | Herbaceous Wetlands | 23,005,862 | 94% | 86% | 88% | 75% |

Nationwide, most crops had within-domain error proportions greater than 50%, which signifies that they were most frequently confused with another crop when mapped incorrectly. Two notable exceptions were alfalfa and fallow/idle cropland, which had within-domain omission error rates of 27 and 35%, respectively. Thus, alfalfa and fallow fields that were incorrectly captured by the CDL were most frequently classified as a non-cropland cover. Alfalfa's within-domain commission error rate was also less than 50%, which suggests that pixels incorrectly mapped as alfalfa in the CDL were most likely to be non-cropland covers on the landscape.

The proportion of within-domain errors for errors of omission for all non-cropland covers were greater than 50%, indicating that misclassified non-cropland covers were most likely to be labeled as another non-crop cover by the CDL. However, aquaculture, developed/open space, and grassland/pasture all had low within-class rates of errors of commission, which indicates that when incorrect, these land covers were frequently mapped in locations that were actually cropland.

*3.4. Spatial Patterns of CDL Accuracy, Confidence, and Certainty*

CDL accuracy for specific crops varied greatly across the U.S. In general, most crop accuracies in 2012 were highest within major cropping regions such as the Corn Belt, Central Plains, and Mississippi Delta (Figure 2a; Appendix A Figure A1). Conversely, crop accuracies were lower along the periphery of these core production zones and in

less dominant agricultural regions of the eastern, southern, and western parts of the U.S. These locations with lower accuracy have a higher prevalence of less common crops (e.g., crops other than corn and soybeans), which are typically mapped less accurately due to more limited reference and training data from FSA and a charter by USDA to focus mapping efforts on major program crops [15,41]. In addition, a greater mixture of crop and non-cropland covers in these areas generates more opportunities for misclassification.

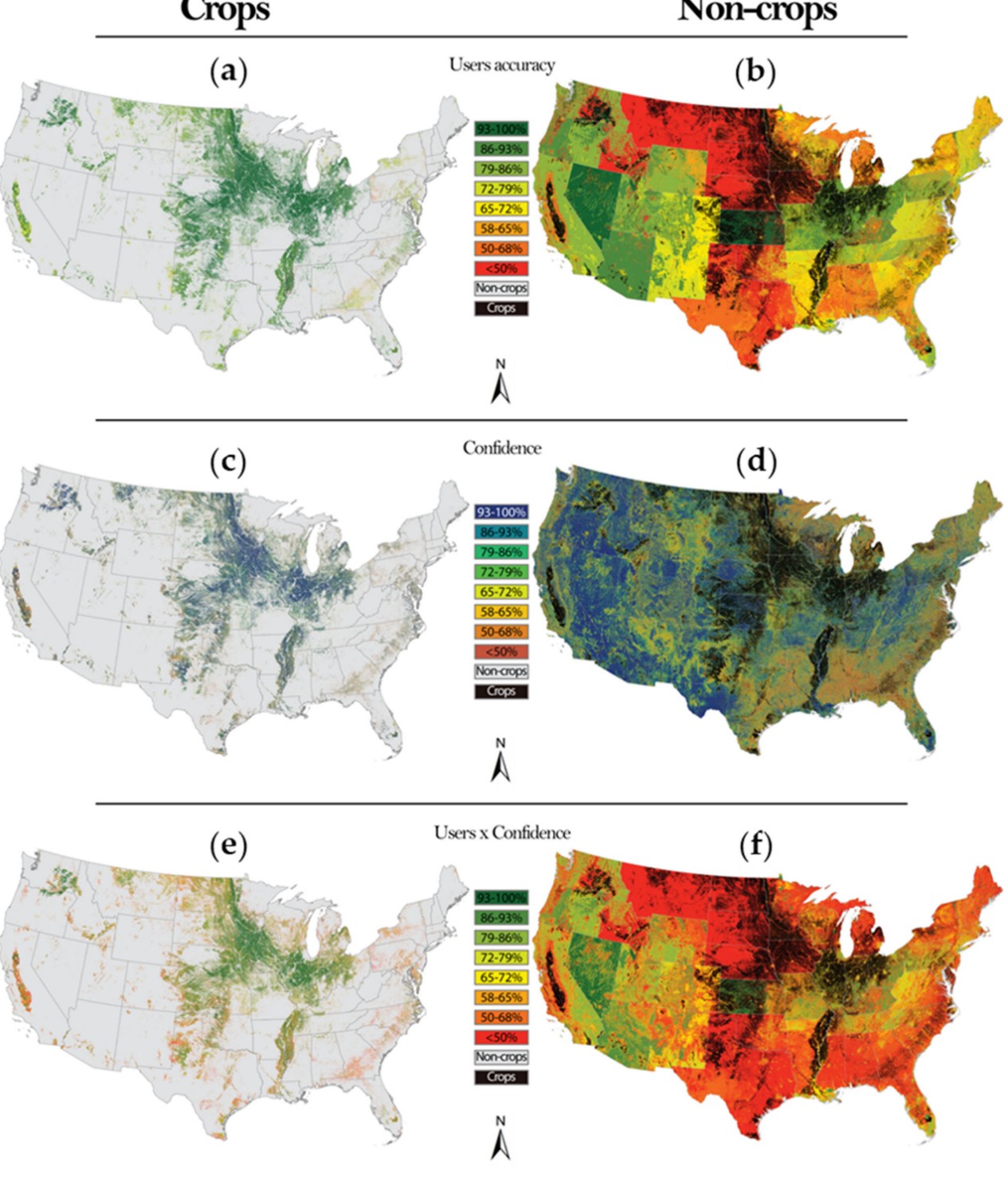

**Figure 2.** Panel of the user's accuracy (**a**,**b**), confidence layer (**c**,**d**), and combined product of user's accuracy and confidence layer (**e**,**f**) delineated for crop (**a**,**c**,**e**) and non-crop (**b**,**d**,**f**) classes of the CDL for 2012.

Non-crop classes had the highest levels of reported disagreement between mapped and reference sources in the northern and southern plains (Figure 2b). Most western states, on the other hand, had a clearer identification of non-cropland cover types, particularly across the vast non-cultivated areas in the region. Mid-Atlantic states and the eastern Corn Belt also contained relatively high non-crop accuracies considering their diverse composition of land cover classes.

The visual inspection of confidence layers suggests that the locations of mixed pixels—map units which fall across two or more land covers—are often mapped with lower confidence than adjacent single cover pixels. For example, in heavily cultivated regions of the country such as Iowa, mixed pixels commonly occur between adjacent fields and along roadways, where they are often the cause of misclassification in the CDL and other remote sensing products [52,53]. In forested regions of the U.S., confidence levels were also low, even across large uninterrupted swaths of forest land cover. In these such locations, the low confidence reflects difficulty by the classification algorithm in delineating the specific type of forest cover—i.e., deciduous, coniferous, mixed forest, or woody wetland.

Regionally, CDL confidence levels are high across the Midwest and west, and lowest in the southeast, northeast, and Great Lakes regions (Figure 2c,d). Within specific regions of similar land cover, there is also variation. For example, in the cultivated region of the Texas panhandle, cotton and corn on the western edge are both mapped with lower confidence, perhaps due to a greater amount of land use change and intermittent cropping patterns in that area. Across the North and South Dakota, crops tend to be consistently mapped with lower confidence the farther west they are located (Appendix A Figure A2).

To extract further insights about the within-class spatial variation of CDL performance, we combined the classifier confidence data with assessed class accuracy into a single measure of CDL certainty. Figure 2e,f shows an example of the combined accuracy x confidence product at the national scale. Integrating pixel resolution spatial variation from the confidence layer into the existing state and class resolution accuracy estimates is particularly applicable to nationwide and multistate analyses since the confidence data have greater continuity among state products.

In addition to helping normalize certainty across regions, the use of both accuracy and confidence information independently or in combination may provide improved insights into local uncertainty. Figure 3 shows an example of an agriculturally intensive region of southern Iowa. Here, accuracy data help demarcate field-sized tracts of land that have low class accuracies (Figure 3a), which are locations that data users may wish to withhold from analyses due to the large uncertainty associated with their classification. Alternatively, the confidence layer captures finer levels of uncertainty due to mixed pixels or other contributors to local uncertainty such as topography or ambiguity among land covers (Figure 3b) but fails to consider the likelihood of the mapped class being incorrect. Considering both accuracy and confidence data (Figure 3c) thus provides insights into multiple dimensions of uncertainty and may be valuable for improving the certitude of mapping and map applications.

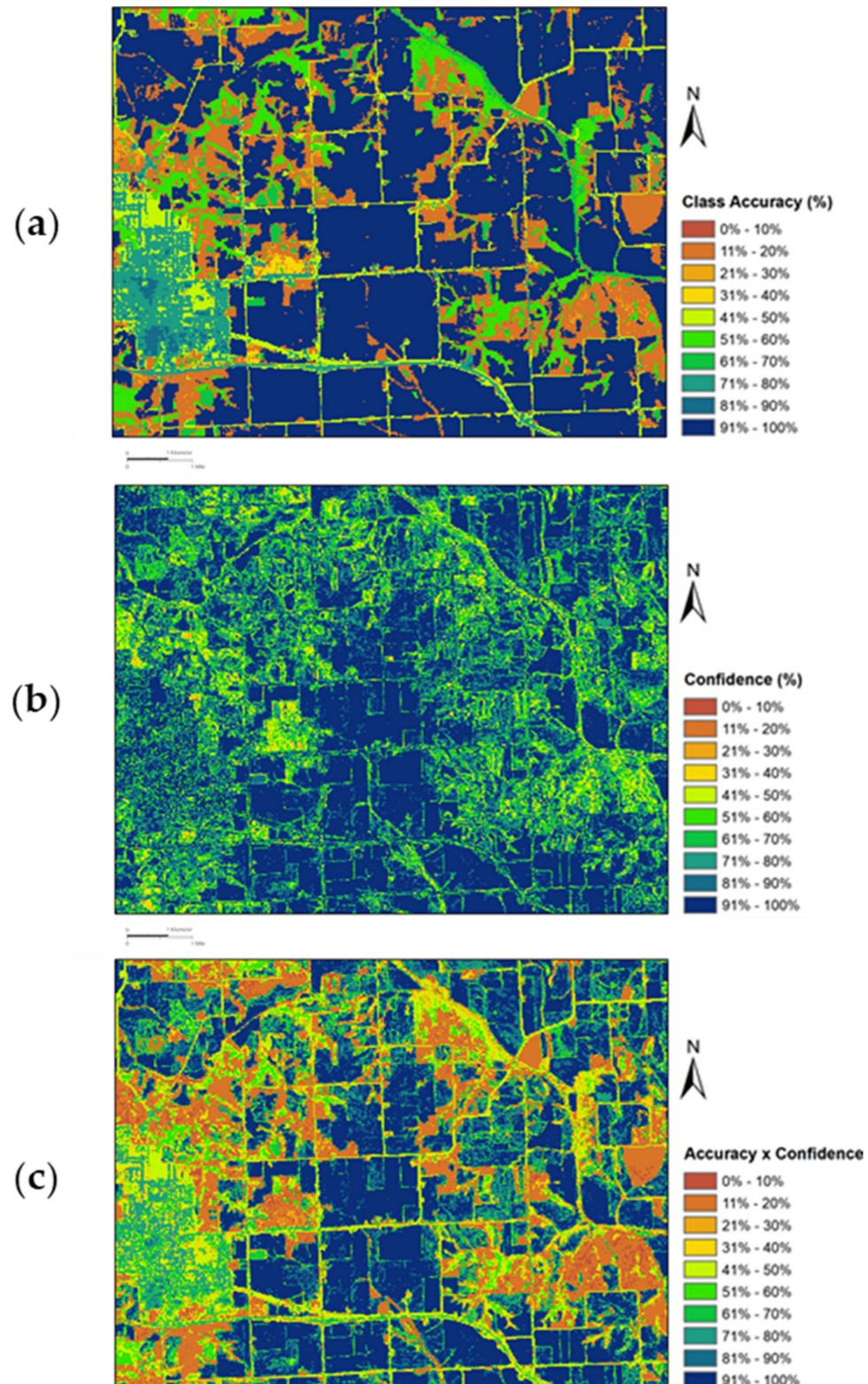

**Figure 3.** Maps of CDL user's accuracy (**a**), confidence levels (**b**), and a combined layer of certainty, shown as the product of accuracy and confidence (**c**).

### 3.5. Measured Biases and Adjusted Crop Area Estimates

Adjusted estimates of crop area informed by map biases can improve upon raw pixel-count area estimates by calibrating them against the reference data used for assessment. Table 6 presents the simple map biases (Equation (12)) and associated adjusted acreage

estimates (Equation (13)) for the 18 largest crop classes for the CDL for which there are also relevant data from official USDA acreage estimates. Given that the CDL represents mid-summer estimates of crop extent, we include NASS data for both planted and harvested areas, as well as the average of these two metrics. For ten of the 16 crops with comparable NASS planted and harvested data, the simple bias-adjusted acreage estimate was closer than the raw pixel-count estimate to the average of NASS planted and harvested areas. As such, the adjusted results provide refined measures of crop area that are independent of (but more consistent with) other acreage estimates such as the NASS Surveys or Census of Agriculture and could be used to complement or replace raw CDL pixel count area estimates in various applications.

**Table 6.** Simple bias and bias-adjusted acreage estimates for major crops for 2012. CDL area represents the summed area of all pixels in the CDL. CDL bias and bias-adjusted acreage were calculated for each crop according to Equations (12) and (13) using the producer's and user's accuracy data of Appendix A Table A2. NASS planted and harvested areas are from the annual NASS acreage report, released on June 29, 2012. Harvested cotton from 2012 October production report. All area values are reported in acres.

| Crop Name | CDL Area | CDL Bias | Bias-Adjusted Acreage | NASS Planted Area | NASS Harvested Area | NASS Ave |
|---|---|---|---|---|---|---|
| Corn | 94,983,301 | 0.43% | 94,572,035 | 96,405,000 | 88,851,000 | 92,628,000 |
| Soybeans | 69,810,086 | −0.03% | 69,829,899 | 76,080,000 | 75,315,000 | 75,697,500 |
| Winter Wheat | 34,784,199 | −0.22% | 34,860,122 | 41,819,000 | 35,023,000 | 38,421,000 |
| Fallow/Idle Cropland | 24,395,076 | −12.24% | 27,382,251 | * 14,145,567 | ** 36,382,032 | n/a |
| Alfalfa | 16,167,152 | −5.52% | 17,059,748 | 19,213,000 | 18,827,000 | 19,020,000 |
| Cotton | 13,114,321 | 1.88% | 12,868,014 | 12,635,000 | 10,443,400 | 11,539,200 |
| Spring Wheat | 12,303,171 | 2.97% | 11,937,985 | 11,995,000 | 11,681,000 | 11,838,000 |
| Sorghum | 6,262,444 | −6.60% | 6,675,868 | 6,210,000 | 5,238,000 | 5,724,000 |
| Dbl Win-Wht/Soybeans | 5,311,121 | 2.85% | 5,159,595 | *** | *** | *** |
| Barley | 2,852,300 | −12.90% | 3,220,316 | 3,678,000 | 3,268,000 | 3,473,000 |
| Rice | 2,671,894 | −1.51% | 2,712,326 | 2,661,000 | 2,640,000 | 2,650,500 |
| Durum Wheat | 1,860,552 | −9.80% | 2,042,794 | 2,203,000 | 2,122,000 | 2,162,500 |
| Dry Beans | 1,743,309 | −6.65% | 1,859,213 | 1,632,700 | 1,573,600 | 1,603,150 |
| Canola | 1,700,926 | −2.23% | 1,738,835 | 1,631,500 | 1,593,100 | 1,612,300 |
| Peanuts | 1,657,438 | −1.42% | 1,680,900 | 1,526,000 | 1,486,000 | 1,506,000 |
| Sunflower | 1,595,069 | −8.79% | 1,735,327 | 1,804,500 | 1,735,400 | 1,769,950 |
| Oats | 1,285,192 | −33.81% | 1,719,707 | 2,746,000 | 1,091,000 | 1,918,500 |
| Sugarbeets | 1,238,159 | −0.55% | 1,244,915 | 1,244,100 | 1,215,900 | 1,230,000 |

\* Estimate of fallow cropland area from the 2012 Census of Agriculture. \*\* Estimate of idle cropland area from the 2012 Census of Agriculture. \*\*\* Double cropped winter wheat/soybean area from the CDL may be added to both CDL soybeans and CDL winter wheat areas to facilitate comparison with NASS estimates for each individual crop.

Assessing the changes in mapped biases over time may also aid in understanding the true dynamic compositions of crops on the landscape. Figure 4 charts the simple bias of four major crops over time. According to the estimates, the mapping of both corn and soybeans by the CDL relative to their reference data have increased only slightly, and in tandem, over time. In contrast, alfalfa has gone from being under-mapped by 12.6% relative to the reference data in 2008 to being under-mapped by only 1.6% in 2016, which marks a considerable change over time. As a result, estimates of alfalfa area based on direct CDL pixel counts could embody a sizeable artificial increase.

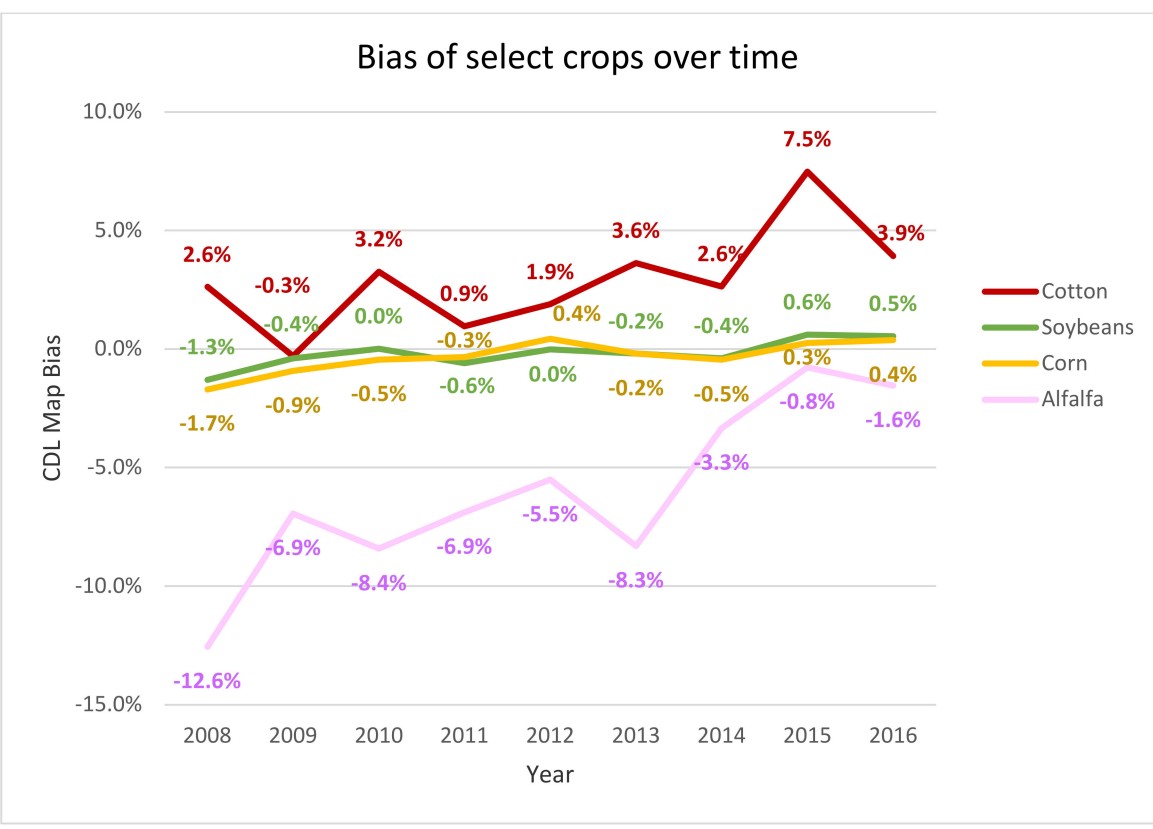

**Figure 4.** Mapping bias of select crops over time. The biases represent the relative over-representation (positive values) or under-representation (negative values) of crops by the CDL in each year according to comparison with the products' reference data.

## 4. Discussion

The Cropland Data Layer currently provides the only annual information on agricultural land use/land cover across the United States that is geographically comprehensive, spatially explicit, and crop specific. Despite its prominent use and application, the accuracy of the CDL had not been well characterized at national scales nor across common aggregated classes. To fill this gap, we derived and analyzed multiple metrics of certainty for the CDL across space and time to better understand its performance and associated implications for measuring LULC and its change.

### 4.1. CDL Performance

Based on nationwide assessment, it is evident that the CDL consistently identifies specific major crops like corn and soybeans with very high accuracy. On the other hand, select land cover classes such as alfalfa and grassland/pasture are captured correctly only about 75% of the time, which reflects the CDL's generally lower performance outside of the major crop classes, a point frequently discussed in state and regional evaluations [35,40].

To accommodate low accuracies, specific classes can be aggregated into broader land cover domains such as cropland or non-cropland. Our results spatially and numerically quantify the effectiveness of this approach and show that across the U.S., cropland areas are mapped correctly by the CDL at least 97% of the time for all years. These findings confirm the CDL's acuity of identification and demonstrate its validity for monitoring cropland locations and associated shifts over time.

Mapping the spatial variation in class accuracies across the United States reveals clear geographic trends and patterns in the CDL's performance. In general, specific crop accuracies are highest within core agricultural areas and among major USDA program crops. Cropland superclass and consolidated cropland accuracies, however, are consistently

high across the country, and further illustrate the value of aggregating to broader domains when attempting to measure land cover across large areas or across all CDL classes, particularly on the margins of major crop zones.

The use of map bias information to adjust area estimates provides a quantitative means to improve crop area calculations based on remote sensing products [10,50]. Similarly, the simplified bias-adjusted approach for estimating crop area reported here improved upon raw CDL pixel-based estimates by correcting for misclassifications and also provides a more comprehensive accounting of cropland than the FSA reference data would provide on its own, since that data source only captures land with crops that participate in FSA programs. Our approach thus combines desirable features of both the CDL and FSA datasets, while remaining independent of other USDA data sources like the NASS surveys or Census of Agriculture that are occasionally used for calibration or comparison.

### 4.2. Improvements over Time

For most metrics, we reported on the performance of the 2012 CDL, although variability exists across years. Overall, CDL accuracy has improved over time, due in part to use of additional satellite input (more sources and more images per year), a more robust classification process (an ensemble decision tree instead of maximum likelihood methodology), and increasing amounts of training data from the FSA and elsewhere [41]. As a result, average class-specific accuracy for all crop classes has improved from 87% in 2008 to 92% in 2016. By 2016, a total of 17 crops were mapped with 90% or higher producer's accuracy, up from just 10 crops in 2008. Aggregate metrics, including consolidated and superclass accuracies for the cropland and non-cropland domains, have also improved. However, the magnitude of their increases is more limited due to their already high performance across time.

The annual changes in performance of the CDL can have important ramifications for CDL-based analyses. If the bias or relative over- or under-mapping of a class changes over time, it can induce false signals of LULC change or skew estimates of crop area change. Lark et al. (2017) explore the implications from the change in total cropland bias and suggest potential solutions [41]. Here, we show that there are also sizable changes in bias for specific crop types. These changes, if disregarded, may influence the results of analyses of those crops over time. For example, unadjusted estimates of the increase in corn acreage following the biofuels boom could be affected by artificial changes in corn mapping across time. However, the magnitude and direction of impact depends on the specific years of analysis and may be counterbalanced by parallel biases in soybeans and other crops. Thus, analyses that focus on the relationship among corn, soybeans, and cropland—or any classes that have experienced synchronized changes in bias—likely remain valid despite potential eccentricities in the underlying data. Nonetheless, it is important to consider the biases of mapped data in applied analyses, particularly when results may influence industry and policymaking.

### 4.3. Implications for Measuring LULC Change

The use of aggregated classes to measure LULC change benefits from the high acuity of the product to detect a broader domain while avoiding challenges of delineating spectrally similar land covers within the same domain. When measuring conversions between cropland and non-cropland, the consolidated classes can thus be used to initially detect change, followed by subsequent identification of the specific land cover or crop planted before and after the conversion [22]. The assessment of crop specificity after detecting change maintains the thematic richness of the original CDL dataset without adversely affecting detection of a conversion between the aggregated domains. In practice, this isolates the known uncertainty in specific class identification and removes it from the change detection process.

Using this two-stage approach, the likelihood that a conversion occurred becomes a function of the highly accurate aggregated classes, whereas the certainty of which specific land cover class preceded and followed a conversion (given that the conversion was

correctly identified) is dependent upon the land cover's specific class accuracy. Thus, for cropland conversion estimates such as Lark et al. (2015) or Morefield et al. (2016), the class accuracies reported in our Table 2 most closely represent the likelihood that a given crop was planted on newly converted land, rather than directly indicate the likelihood that a conversion occurred [22,54].

The challenges of mapping less-common specific crops and the ease of mapping aggregate cropland have additional implications for CDL-based applications. For example, it might be argued that the CDL is more appropriate for detecting broad land use changes (e.g., conversion between cropland and non-cropland) than for identifying nuanced changes among specific crops (e.g., identifying crop rotations) unless the focus of rotations remains on major crop types [18,19,55]. Crop-specific applications should also consider each class's prevalence and accuracy and how such factors may influence results.

Our findings can also be used to guide how specific crops should be treated within analyses. Alfalfa, for example, is often cited as a problem crop due to its semi-perennial nature, spectral similarity to non-cropland covers, and occasional interplanting within mixed species hay and pasture. The crop was incorrectly mapped in non-cultivated areas 14% of the time in 2012. By 2016, this superclass error rate dropped to just 8%. From a producer's perspective, alfalfa was mapped as a non-cultivated land cover 20% of the time in 2012, but this error rate dropped to 8% by 2016. Overall, the lower superclass accuracies for alfalfa relative to other crops reinforce precautions of past analyses, such as the exclusion by Morefield et al. (2016) of all non-crop to alfalfa conversions from their change analysis and the exclusion by Lark et al. (2015) of grassland-to-alfalfa conversion. The relative within-domain error rates (Table 4) further highlight the challenge of including alfalfa in the cropland domain, since the crop is more frequently confused with non-cropland covers than with other crop classes. However, the latest improvements in alfalfa accuracy suggest that analyses of more recent CDL data may want to consider including the forage crop in their analyses.

Visual mapping of specific and aggregate accuracies can help users identify hotspots and problem areas within the country and understand how they vary across space and time. Coupling accuracy data with its spatial location on the landscape thus offers opportunities unafforded by the nonspatial structure of the NASS metadata tables and confusion matrices for each state and year. For example, rather than excluding entire land cover classes from analyses, such as the exclusions of alfalfa by Morefield et al. (2016) and Lark et al. (2015), the spatial mapping of the accuracy of individual classes would allow the empirical removal of just those pixels with low mapped accuracy in certain state–year combinations, while retaining those with a higher likelihood of being correct. The value of this spatial approach is greatest in analyses that consider multiple years of CDL data, where the number of state, class, and year combinations is multiplicative. For example, for an assessment of change between two years, there are typically over a million unique combinations of state and class pairs, each with its own likelihood of being correct (e.g., 50 classes times 40 states for year one multiplied by 50 classes times 40 states for the second year yields four million combinations). The manual selection of which specific LULC class combinations to include or exclude based on accuracy thus becomes intractable, whereas the spatial accuracy maps can be used to easily select only those combinations that meet a quantitative accuracy threshold.

The integration of confidence layer data with assessed accuracy data may also improve spatial insights. For example, in many CDL-based change detection analyses, post-classification processes such as spatial filters and minimum mapping units have been used to indiscriminately remove areas of apparent change that are likely falsely mapped due to mixed pixels or misclassifications. Alternatively, accuracy and confidence data could be used to set a threshold of certitude below which any identified potential change is flagged for removal. Probability information from the remote sensing process has previously been used to improve vector-based detection of land cover change using unclassified Landsat data [48]. Here, we suggest that confidence information from the remote sensing process

could similarly help improve the *post-classification* detection of LULC change using land cover products. While we have not quantified the impact of such an approach, it has since been used in other studies to set a higher threshold of certainty for change detection [56].

Confidence layer data could also be used in concert with accuracy information to spatially allocate error adjustments. For example, here we modified area estimates for each crop using an accuracy-derived indicator of bias (Table 6). However, such area adjustments typically do not spatially correct pixels on the map, unless this issue of reconciliation is specifically addressed [57]. To help achieve this reconciliation in post-classification environments, confidence data could similarly be used to select the pixels with the lowest confidence as candidates for reclassification. For example, if the CDL overestimated corn area by 500 pixels in a given state, the 500 pixels of corn with the lowest confidence could be removed to make a spatially explicit, bias-adjusted map of corn that was consistent with the reference data estimates of area.

*4.4. Limitations, Representativeness, and Uncertainty of Results*

The class consolidation techniques described here do not modify the underlying performance of the remote sensing product, but rather improve the representativeness of the accuracy at which the product maps aggregate domains. Of note, aggregating classes improves accuracy by lowering the product's thematic resolution or specificity—thus improvements are made by accommodating errors rather than by correcting them. The greatest benefits are therefore achieved when the thematic resolution of the product matches the desired application. When using aggregated remote sensing products in applications, it is important to quantify these associated changes in accuracy so that the reported metrics and critiques reflect the actual data used.

There may also be variation in the representativeness of the CDL's reported accuracy statistics. The FSA reference data used to assess the CDL are not based on a probabilistic sample, but rather on an availability approach, with the majority coming from 10 key USDA program crops. As a result, the reported consolidated class accuracies are most representative for those crops, and less characteristic for specialty crops and non-crop covers. Similarly, the distribution of crop sample data across geographic regions are in some places disproportionate to the amount of crop produced there. Therefore, the accuracies of certain regions are more reliable than others due to differing levels of reference data available for assessment.

To maintain the highest level of representativeness while calculating national average crop accuracies, we weighted the accuracy of each crop in each state by the total acreage of that crop in that state. For example, Iowa produced 14% of all corn in the nation in 2012; thus, its accuracy was weighted to contribute 14% of the national accuracy for corn. An alternative method for calculating nationwide accuracies is to sum all national reference observations without regard to spatial distributions of the data, and such an approach has recently been implemented by NASS to report nationwide accuracies for select years in the online CDL metadata [26]. Here, we choose to area-weight by class prevalence, such that the nationwide estimates reflect that of a pixel selected at random and are unskewed by nonrepresentatively sampled reference data.

Uncertainty can also stem from errors in the reference data or a mismatch between reference and evaluated data. For example, the FSA CLU classifications of grasslands are often inconsistent across states and time, and occasionally they do not align with CDL land cover designations. Thus, analysts at NASS make a judgement for each state and year on how to best utilize the FSA data for training and assessing accuracy. Discrepancies in how the FSA data are reported and incorporated can thus occasionally lead to apparent differences in error rates across states and years, when in reality the inconsistencies between the CDL and the landscape are much smaller. Similarly, errors exist in the NLCD data used for training and assessing non-crop areas of the CDLs, which in turn affect their production and assessment. It is possible that some CDL non-crop classes are more correct than the associated NLCD classes on which they are based and evaluated, given that the CDL is

updated and improved annually, it includes exclusive confidential FSA training data, and it generates higher accuracies for cultivated areas. Thus, the reported non-crop accuracies of the CDL (based on comparison with the NLCD) may underestimate the true performance of those CDL classes.

## 5. Conclusions

The CDL is a powerful and unrivaled tool for the exploration of agricultural landscapes and is poised to remain the premier remotely sensed agricultural LULC map in the U.S. due to its annual availability, crop-specific detail, and exclusive access to expansive and robust ground-based reference datasets from the USDA. We show that the CDL identifies major crops and certain land covers with high accuracy across the U.S., and that this ability holds true for all years of nationwide data coverage. Our findings also confirm that the CDL exhibits extremely high acuity at discerning the aggregated classes of cropland and non-cropland across spatial and thematic scales. Explicitly considering the bias within specific classes and incorporating confidence layer data provide two additional opportunities to further improve CDL performance and its use in LULC change assessments and other applications.

While the original CDL dataset can indeed provide challenges for applications that are beyond its original intent of mapping annual crop locations, it is the responsibility of its users to apply the data in ways that do not compromise results. The CDL's consistent and reliable performance in mapping crops and cropland nationwide and across time clearly demonstrates that many of the critiques and concerns regarding the underlying accuracy of the product are unfounded or dissipate when thoroughly assessed at appropriate scales. Furthermore, the substantial uncertainty and resource costs of alternative methods for monitoring crops and croplands, such as through ground surveys or air photo interpretations, underscores the need for approaches that can systematically identify continental scale LULC change in an automated, reproducible, and verifiable manner. While many products based on remote sensing seek to fill this gap, the CDL is a dataset proven to be well suited for the task. When used appropriately, the CDL is a valid and indispensable tool for studying LULC and a crucial asset for monitoring contemporary cropland dynamics across the United States.

**Author Contributions:** Conceptualization, T.J.L. and H.K.G.; formal analysis, T.J.L. and I.H.S.; writing—original draft preparation, T.J.L.; writing—review and editing, T.J.L., I.H.S., and H.K.G. All authors have read and agreed to the published version of the manuscript.

**Funding:** This material is based upon work supported in part by the National Science Foundation (DRL-1713110) and the Great Lakes Bioenergy Research Center, U.S. Department of Energy, Office of Science, Office of Biological and Environmental Research (DE-SC0018409).

**Data Availability Statement:** All results and data including those for additional years of the CDL have been archived online via Zenodo and are available at https://doi.org/10.5281/zenodo.4579863 (accessed on 1 January 2021).

**Acknowledgments:** Special thanks to Rick Mueller, Dave Johnson, and Patrick Willis at USDA NASS for their helpful discussions and for providing CDL confidence data. Thanks also to Meghan Salmon for her insights and initial help exploring consolidated crop accuracies using the CDL supermatrices and to Carol Barford and Volker Radeloff for their feedback and suggestions of analyses from early discussions of this manuscript. Thanks to George Allez for editing.

**Conflicts of Interest:** The authors declare no conflict of interest.

# Appendix A

**Table A1.** List of CDL codes and class names and whether they were included in the cropland or non-cropland domain in the analyses of superclass and consolidated class accuracies. Domain delineations follow that of Lark et al. (2015) based on original NASS distinctions [16,22].

| Cropland | | | | | | Non-Cropland | |
|---|---|---|---|---|---|---|---|
| ID | Class Name | ID | Class Name | ID | Class Name | ID | Class Name |
| 1 | Corn | 48 | Watermelons | 216 | Peppers | 37 | Other Hay/Non Alfalfa |
| 2 | Cotton | 49 | Onions | 217 | Pomegranates | | |
| 3 | Rice | 50 | Cucumbers | 218 | Nectarines | 63 | Forest |
| 4 | Sorghum | 51 | Chickpeas | 219 | Greens | 64 | Shrubland |
| 5 | Soybeans | 52 | Lentils | 220 | Plums | 65 | Barren |
| 6 | Sunflower | 53 | Peas | 221 | Strawberries | 81 | Clouds/No Data |
| 10 | Peanuts | 54 | Tomatoes | 222 | Squash | 82 | Developed |
| 11 | Tobacco | 55 | Caneberries | 223 | Apricots | 83 | Water |
| 12 | Sweet Corn | 56 | Hops | 224 | Vetch | 87 | Wetlands |
| 13 | Pop or Orn Corn | 57 | Herbs | 225 | Dbl Crop WinWht/Corn | 88 | Nonag/Undefined |
| 14 | Mint | 58 | Clover/Wildflowers | 226 | Dbl Crop Oats/Corn | 92 | Aquaculture |
| 21 | Barley | 59 | Sod/Grass Seed | 227 | Lettuce | 111 | Open Water |
| 22 | Durum Wheat | 60 | Switchgrass | 229 | Pumpkins | 112 | Perennial Ice/Snow |
| 23 | Spring Wheat | 61 | Fallow/Idle | 230 | Dbl Crop Lettuce/Durum Wht | 121 | Developed/Open Space |
| 24 | Winter Wheat | 66 | Cherries | 231 | Dbl Crop Lettuce/Cantaloupe | 122 | Developed/Low Intensity |
| 25 | Other Small Grains | 67 | Peaches | 232 | Dbl Crop Lettuce/Cotton | 123 | Developed/Med Intensity |
| 26 | Dbl WinWht/Soy | 68 | Apples | 233 | Dbl Crop Lettuce/Barley | 124 | Developed/High Intensity |
| 27 | Rye | 69 | Grapes | 234 | Dbl Crop Durum Wht/Sorghum | 131 | Barren |
| 28 | Oats | 70 | Christmas Trees | 235 | Dbl Crop Barley/Sorghum | 141 | Deciduous Forest |
| 29 | Millet | 71 | Other Tree Crops | 236 | Dbl Crop WinWht/Sorghum | 142 | Evergreen Forest |
| 30 | Speltz | 72 | Citrus | 237 | Dbl Crop Barley/Corn | 143 | Mixed Forest |
| 31 | Canola | 74 | Pecans | 238 | Dbl Crop WinWht/Cotton | 152 | Shrubland |
| 32 | Flaxseed | 75 | Almonds | 239 | Dbl Crop Soybeans/Cotton | | |
| 33 | Safflower | 76 | Walnuts | 240 | Dbl Crop Soybeans/Oats | 176 | Grassland/Pasture |
| 34 | Rape Seed | 77 | Pears | 241 | Corn/Soybeans | | |
| 35 | Mustard | 204 | Pistachios | 242 | Blueberries | 190 | Woody Wetlands |
| 36 | Alfalfa | 205 | Triticale | 243 | Cabbage | 195 | Herbaceous Wetlands |
| 38 | Camelina | 206 | Carrots | 244 | Cauliflower | | |
| 39 | Buckwheat | 207 | Asparagus | 245 | Celery | | |
| 41 | Sugarbeets | 208 | Garlic | 246 | Radishes | | |
| 42 | Dry Beans | 209 | Cantaloupes | 247 | Turnips | | |
| 43 | Potatoes | 210 | Prunes | 248 | Eggplants | | |
| 44 | Other Crops | 211 | Olives | 249 | Gourds | | |
| 45 | Sugarcane | 212 | Oranges | 250 | Cranberries | | |
| 46 | Sweet Potatoes | 213 | Honeydew Melons | 254 | Dbl Crop Barley/Soybeans | | |
| 47 | Misc Vegs and Fruits | 214 | Broccoli | | | | |

**Table A2.** Nationwide area, producer's accuracy, and user's accuracy for each crop type in the 2012 CDL. Sorted in order of descending producer's accuracy.

| CDL ID | Crop Name | CDL Acreage | Producer's Accuracy | User's Accuracy |
|---|---|---|---|---|
| 45 | Sugarcane | 1,026,752 | 96.52% | 94.44% |
| 3 | Rice | 2,671,894 | 95.54% | 97.01% |
| 1 | Corn | 94,983,301 | 95.23% | 94.82% |
| 5 | Soybeans | 69,810,086 | 93.82% | 93.85% |
| 41 | Sugarbeets | 1,238,159 | 93.67% | 94.18% |
| 31 | Canola | 1,700,926 | 93.51% | 95.64% |
| 24 | Winter Wheat | 34,784,199 | 92.18% | 92.38% |
| 2 | Cotton | 13,114,321 | 91.06% | 89.39% |
| 75 | Almonds | 1,155,344 | 91.04% | 91.56% |
| 250 | Cranberries | 36,040 | 91.02% | 95.23% |
| 23 | Spring Wheat | 12,303,171 | 89.47% | 86.89% |
| 212 | Oranges | 1,019,334 | 89.24% | 91.45% |
| 54 | Tomatoes | 353,534 | 89.24% | 89.60% |
| 51 | Chickpeas | 1838 | 89.19% | 84.44% |
| 43 | Potatoes | 1,083,450 | 88.98% | 92.66% |
| 69 | Grapes | 1,136,718 | 87.39% | 89.89% |
| 26 | Dbl Crop WinWht/Soybeans | 5,311,121 | 86.70% | 84.30% |
| 230 | Dbl Crop Lettuce/Durum Wht | 39,776 | 86.08% | 80.01% |
| 68 | Apples | 444,242 | 85.67% | 88.41% |
| 56 | Hops | 24,903 | 84.53% | 96.44% |
| 6 | Sunflower | 1,595,069 | 84.09% | 92.20% |
| 10 | Peanuts | 1,657,438 | 81.17% | 82.33% |
| 42 | Dry Beans | 1,743,309 | 79.97% | 85.66% |
| 204 | Pistachios | 201,944 | 78.50% | 85.69% |
| 46 | Sweet Potatoes | 84,332 | 77.54% | 87.22% |
| 4 | Sorghum | 6,262,444 | 77.43% | 82.91% |
| 77 | Pears | 28,048 | 77.36% | 80.67% |
| 36 | Alfalfa | 16,167,152 | 75.40% | 79.81% |
| 245 | Celery | 2460 | 74.95% | 93.43% |
| 76 | Walnuts | 341,480 | 74.80% | 79.49% |
| 52 | Lentils | 388,352 | 74.57% | 82.45% |
| 22 | Durum Wheat | 1,860,552 | 73.30% | 81.26% |
| 49 | Onions | 139,769 | 72.90% | 78.67% |
| 66 | Cherries | 199,450 | 72.70% | 78.60% |
| 211 | Olives | 45,218 | 72.58% | 90.34% |
| 21 | Barley | 2,852,300 | 72.41% | 83.14% |
| 247 | Turnips | 1990 | 72.37% | 79.65% |
| 53 | Peas | 774,135 | 72.14% | 83.45% |
| 208 | Garlic | 17,233 | 71.20% | 84.66% |

**Table A2.** *Cont.*

| CDL ID | Crop Name | CDL Acreage | Producer's Accuracy | User's Accuracy |
|---|---|---|---|---|
| 61 | Fallow/Idle Cropland | 24,395,076 | 69.29% | 78.96% |
| 32 | Flaxseed | 284,228 | 68.10% | 81.77% |
| 59 | Sod/Grass Seed | 797,216 | 68.00% | 82.93% |
| 57 | Herbs | 104,376 | 67.07% | 86.46% |
| 14 | Mint | 8429 | 67.00% | 77.65% |
| 50 | Cucumbers | 32,698 | 65.44% | 78.26% |
| 12 | Sweet Corn | 301,474 | 65.35% | 80.95% |
| 244 | Cauliflower | 1956 | 64.25% | 79.31% |
| 226 | Dbl Crop Oats/Corn | 109,775 | 63.82% | 62.71% |
| 234 | Dbl Crop Durum Wht/Sorghum | 4095 | 63.24% | 66.43% |
| 47 | Misc Vegs and Fruits | 47,159 | 62.89% | 78.30% |
| 225 | Dbl Crop WinWht/Corn | 402,067 | 61.81% | 69.33% |
| 71 | Other Tree Crops | 68,927 | 61.69% | 75.22% |
| 27 | Rye | 453,504 | 61.47% | 72.91% |
| 254 | Dbl Crop Barley/Soybeans | 113,764 | 61.24% | 78.16% |
| 72 | Citrus | 139,758 | 60.68% | 81.33% |
| 33 | Safflower | 148,336 | 59.74% | 80.07% |
| 11 | Tobacco | 112,733 | 59.62% | 79.97% |
| 232 | Dbl Crop Lettuce/Cotton | 7770 | 58.53% | 69.78% |
| 213 | Honeydew Melons | 6430 | 58.09% | 75.87% |
| 231 | Dbl Crop Lettuce/Cantaloupe | 3833 | 57.97% | 85.54% |
| 242 | Blueberries | 90,911 | 57.70% | 74.20% |
| 248 | Eggplants | 357 | 57.69% | 68.18% |
| 227 | Lettuce | 28,621 | 57.45% | 66.98% |
| 58 | Clover/Wildflowers | 146,851 | 57.21% | 70.80% |
| 209 | Cantaloupes | 18,325 | 57.00% | 72.44% |
| 217 | Pomegranates | 20,652 | 56.79% | 76.84% |
| 216 | Peppers | 19,796 | 55.46% | 67.81% |
| 207 | Asparagus | 19,258 | 54.93% | 78.11% |
| 74 | Pecans | 398,572 | 53.68% | 83.55% |
| 29 | Millet | 457,674 | 53.43% | 64.84% |
| 221 | Strawberries | 43,438 | 52.63% | 80.70% |
| 39 | Buckwheat | 22,586 | 52.11% | 78.32% |
| 246 | Radishes | 10,175 | 50.75% | 70.24% |
| 235 | Dbl Crop Barley/Sorghum | 12,071 | 49.65% | 50.19% |
| 67 | Peaches | 53,255 | 49.19% | 68.69% |
| 35 | Mustard | 32,734 | 48.15% | 78.68% |
| 241 | Dbl Crop Corn/Soybeans | 16,998 | 48.07% | 75.59% |
| 60 | Switchgrass | 10,684 | 47.62% | 56.33% |
| 220 | Plums | 53,436 | 46.92% | 65.53% |

**Table A2.** *Cont*.

| CDL ID | Crop Name | CDL Acreage | Producer's Accuracy | User's Accuracy |
|---|---|---|---|---|
| 55 | Caneberries | 11,633 | 46.19% | 85.35% |
| 206 | Carrots | 42,670 | 45.93% | 70.76% |
| 229 | Pumpkins | 23,094 | 43.87% | 72.29% |
| 70 | Christmas Trees | 65,800 | 43.65% | 75.04% |
| 243 | Cabbage | 18,368 | 43.03% | 59.38% |
| 38 | Camelina | 4977 | 42.94% | 69.91% |
| 214 | Broccoli | 11,202 | 41.89% | 63.04% |
| 28 | Oats | 1,285,192 | 41.18% | 62.21% |
| 238 | Dbl Crop WinWht/Cotton | 324,242 | 41.14% | 70.23% |
| 48 | Watermelons | 37,670 | 40.78% | 62.93% |
| 219 | Greens | 15,028 | 40.62% | 54.26% |
| 13 | Pop or Orn Corn | 120,463 | 40.33% | 91.15% |
| 223 | Apricots | 3760 | 39.37% | 71.61% |
| 222 | Squash | 20,832 | 37.05% | 61.87% |
| 237 | Dbl Crop Barley/Corn | 37,530 | 36.55% | 70.59% |
| 236 | Dbl Crop WinWht/Sorghum | 386,258 | 34.26% | 60.55% |
| 224 | Vetch | 4595 | 33.12% | 69.06% |
| 44 | Other Crops | 171,449 | 32.93% | 63.82% |
| 205 | Triticale | 156,684 | 32.74% | 67.26% |
| 218 | Nectarines | 2589 | 32.16% | 70.51% |
| 25 | Other Small Grains | 5008 | 28.18% | 73.07% |
| 34 | Rape Seed | 3211 | 23.92% | 58.74% |
| 239 | Dbl Crop Soybeans/Cotton | 7388 | 20.90% | 66.57% |
| 240 | Dbl Crop Soybeans/Oats | 17,928 | 19.50% | 62.42% |
| 30 | Speltz | 2811 | 16.32% | 60.80% |
| 249 | Gourds | 150 | 10.00% | 100.00% |

**Table A3.** Accuracy of CDL-derived consolidated cropland and non-cropland classifications for each U.S. state or multistate region for 2012. Results calculated according to Equation (4) and consolidated according to Appendix A Table A1.

| State | Cropland | | Non-Cropland | |
|---|---|---|---|---|
| | Producer's Accuracy | User's Accuracy | Producer's Accuracy | User's Accuracy |
| AL | 84% | 93% | 98% | 94% |
| AR | 97% | 100% | 99% | 89% |
| AZ | 91% | 97% | 99% | 95% |
| CA | 96% | 98% | 99% | 93% |
| CO | 93% | 98% | 98% | 90% |
| CT_MA_ME_NH_RI_VT | 87% | 93% | 100% | 99% |
| DE_MD_NJ | 93% | 94% | 98% | 96% |
| FL | 89% | 95% | 98% | 92% |
| GA | 86% | 91% | 98% | 93% |
| IA | 97% | 99% | 95% | 77% |
| ID | 93% | 96% | 99% | 95% |
| IL | 98% | 97% | 92% | 95% |
| IN | 98% | 97% | 94% | 96% |
| KS | 97% | 99% | 99% | 95% |
| KY | 92% | 99% | 97% | 84% |

**Table A3.** *Cont.*

| State | Cropland | | Non-Cropland | |
|---|---|---|---|---|
| | Producer's Accuracy | User's Accuracy | Producer's Accuracy | User's Accuracy |
| LA | 95% | 98% | 98% | 90% |
| MI | 96% | 95% | 95% | 90% |
| MN | 98% | 98% | 97% | 93% |
| MO | 98% | 96% | 97% | 98% |
| MS | 94% | 98% | 98% | 92% |
| MT | 91% | 95% | 99% | 98% |
| NC | 91% | 95% | 97% | 93% |
| ND | 95% | 98% | 97% | 90% |
| NE | 97% | 100% | 95% | 67% |
| NM | 88% | 98% | 99% | 87% |
| NV | 88% | 96% | 100% | 99% |
| NY | 86% | 88% | 98% | 95% |
| OH | 96% | 97% | 96% | 95% |
| OK | 97% | 100% | 96% | 62% |
| OR | 93% | 97% | 98% | 93% |
| PA | 83% | 82% | 98% | 96% |
| SC | 86% | 92% | 98% | 94% |
| SD | 95% | 97% | 98% | 97% |
| TN | 95% | 99% | 98% | 90% |
| TX | 92% | 100% | 96% | 56% |
| UT | 90% | 98% | 99% | 94% |
| VA_WV | 92% | 94% | 99% | 98% |
| WA | 97% | 98% | 100% | 97% |
| WI | 95% | 97% | 93% | 84% |
| WY | 88% | 98% | 99% | 93% |

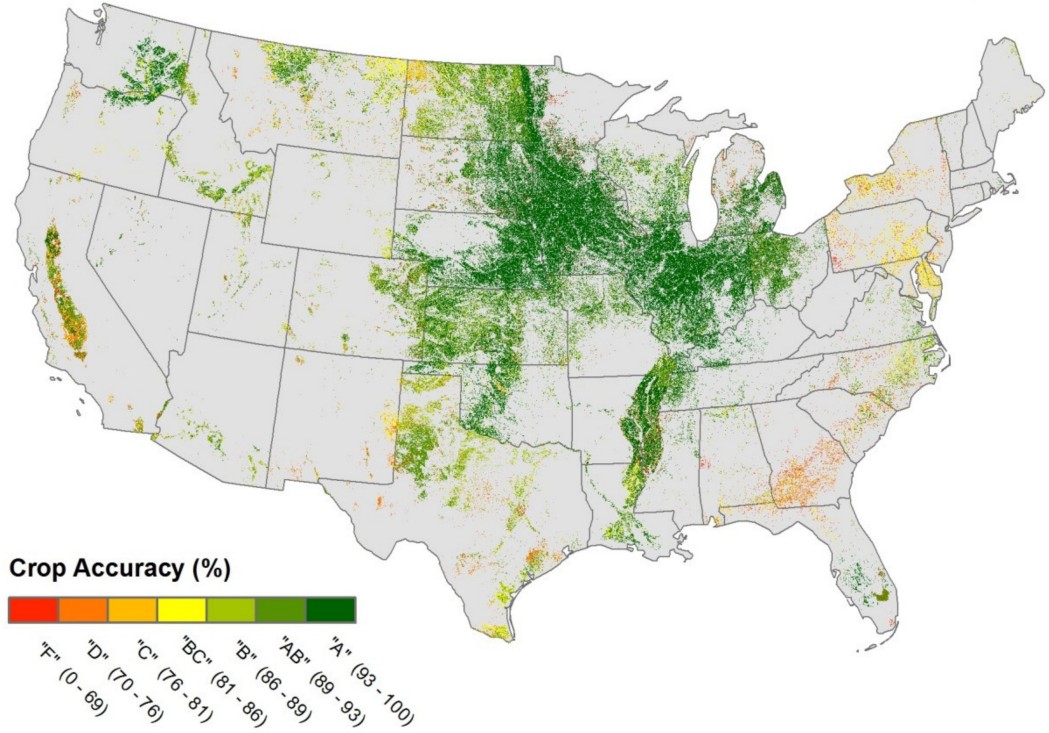

**Figure A1.** Map of 2012 state level user's accuracies for specific crop classes of the CDL for the conterminous U.S. Data from USDA NASS (2016) based on the comparison of CDL with FSA reference data for crop classes. An arbitrary grading scale of "A"–"F" was assigned to accuracy intervals to help users easily identify where the CDL crop map excels versus where additional caution may be warranted.

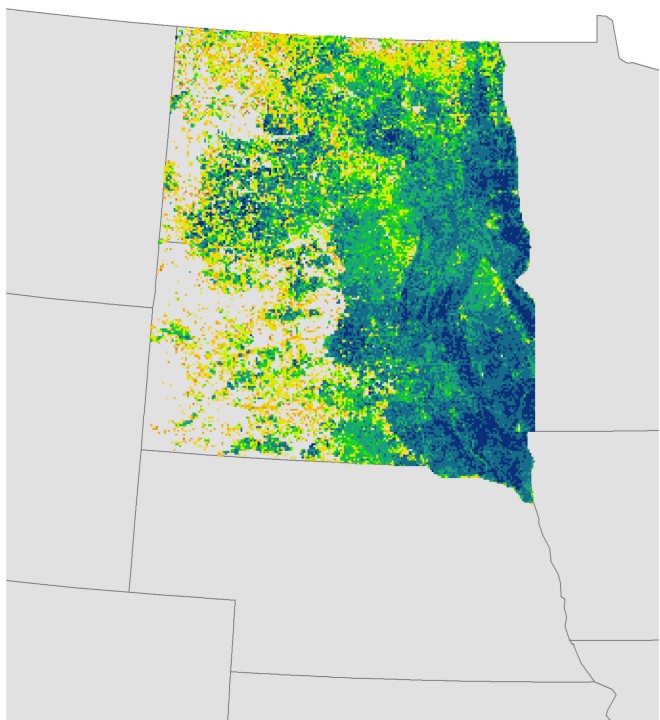

**Figure A2.** Confidence of pixels mapped as corn in the 2012 CDL. Within a specific state, there can be large spatial variation in the degree of certainty with which specific crops are mapped. In South Dakota and North Dakota, corn is mapped more confidently in the eastern parts of the states (dark blue), where the crop is more prevalent, and is mapped less confidently (green to yellow) as one moves westward and the crop becomes less prominent.

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
