# Peer review of "Accuracy, Bias, and Improvements in Mapping Crops and Cropland across the United States Using the USDA Cropland Data Layer"

_remotesensing, doi:10.3390/rs13050968_

Round 1

Reviewer 1 Report

The authors presented a comprehensive assessment of CDL performance at a national scale for mapping crops and cropland. In general, the ms was well-written, and I enjoy reading it.

My major concern is that novelty level of this ms is not very high. Although the authors proposed a new way (perhaps, a good way) to characterize CDL performance at national scale, there are no real “new” findings in the study. Good performance on crop categories, especially for major crops / agricultural states, is something we can always expect from CDL. It is indeed just a typical property of any land cover classification — major classes always have higher classification accuracy. Good separation between crop and non-crop is also expected for most land cover products (including CDL). Similarly, aggregating land cover classes would always boost accuracy level of any land cover product.

Minor comments: figures look a bit fuzzy. Perhaps, the authors can increase their DPI (and make larger labels)  

Reviewer 2 Report

This manuscript provides an interesting assessment of the accuracy of the USDA Cropland Data Layer, which is a very important source of cropland data in the US. The manuscript generally provides a clear explanation of the methods implemented and an appropriately detailed description of results.

Major Comments

  1. It would be helpful if some additional information would be provided regarding the reference data, perhaps in the text at approximately Lines 130-L133. For example, consider providing additional information about the FSA data (what is the source of the information, how is land cover spatially represented, how are locations selected). Later in the manuscript it is stated that the FSA data are not selected via a probability sampling design. That information should be provided here. Similarly, the authors mentions at L287-289 that NLCD is another classification of land cover, which is generally different from what is usually considered “reference data” (i.e., NLCD is not independent, high quality interpretation of a sample of reference locations). As a last detail, NLCD is not an annual product so that indicates NLCD data were not available for all years. So some years are assessed using just FSA? The explanation of Methods would be improved if the information regarding the reference data used as the basis of the comparisons would be consolidated in one place in the text.
  2. In my opinion, the “bias adjusted area” results are the weakest component of the entire manuscript. First, it is questionable how these area estimates were obtained. Equation (12) is not how “bias adjusted” estimation was described and implemented in Olofsson et al. (2013, 2014), which is cited as the source for how bias adjusted area is computed. Olofsson et al.’s (2013) equation (2’) is a stratified estimator of area and that formula bears no resemblance to equation (13) in the manuscript. Equation (12) is not the correct estimator of the bias of the map – the correct estimator of bias would subtract the off-diagonal entries of the error matrix, or alternatively, would take the total area of the class as identified from the map and subtract the total area of the class as determined from the reference data (see for example the recent review article by Stehman and Foody, 2019, Key issues in rigorous accuracy assessment of land cover products, Remote Sensing of Environment 231, 111191). Lastly, the biased adjusted estimators are intended for use with a probability sampling design, so the application of that approach to the data used in this study is much less compelling given that the reference sample data are not from a probability sampling design and some of the reference data are another classification (NLCD). The primary value of the manuscript is the accuracy results. The so-called adjusted area estimates are far less useful.
  3. Lines 658-L661: The two sentences that cite reference [38] do not make sense. The reason reference data are not used in place of the mapped product is because we do not have full coverage reference data, we only have a sample, often only a very small portion of the region of interest. Secondly, if the reference sample is selected via a probability sampling design, as is the good practice recommendation specified in Olofsson et al. (2014), the sample is by definition representative, so the statement in the manuscript about compromising reliability and representativeness is confusing (and probably incorrect). The whole purpose of obtaining a sample of reference data is so that an intensive effort can be invested to obtain the correct reference class labels on a relatively small (but probabilistically selected) subset of the entire region. I do not question the value of the FSA data to provide information useful to evaluate the CDL products. But please remove the two sentences that cite [38] because those sentences express invalid ideas.
  4. Table 1 is ambiguous regarding the spatial unit of the assessment (i.e., the unit for which it is decided if the map class is correct). For example, the User’s example indicates the spatial unit is a pixel, whereas the Producer’s example indicates that the assessment unit is a field. It may be that data are available at the field level. So is a map field compared to a reference (“true”) field and somehow agreement between area is quantified? Or is the comparison at the individual pixel level and then the pixel data aggregated to the state or other region?
  5. Equation (1) assumes simple random sampling of pixels, with no stratification. It would be more transparent to state that the data are being treated as a simple random sample. The authors mention that the FSA data are not collected by a probability sampling design, and that is reasonable and provides transparency about the data. It would still be relevant to state that the data are treated as if the design had been simple random sampling for the purpose of producing the estimates.
  6. I am not certain I understand the computation of superclass accuracy. For each specific crop x, we would have two values of superclass accuracy, one for the domain cropland and one for the domain noncropland? So for example for the cropland domain and the specific crop class of corn, the numerator in equation (2) would be the number of pixels mapped as corn that were in the cropland domain, and the denominator would be the number of pixels mapped as corn – this would be superclass user’s accuracy.

Minor Comments

  1. L240-241: This statement does not make much sense because we have no definition of “utility” so we have no criteria for determining utility of the formula
  2. The accuracy results reported in Tables 2 and 3 would be much simpler to evaluate if the accuracy values were rounded to whole numbers. Differences in accuracy of 1% are not practically meaningful (e.g., would a user really conclude anything differently if accuracy is 64% versus 65% or 81% versus 80%?). Therefore, it is not informative to have the decimal place reported. Table 4 goes one step worse by reporting accuracy to two decimal places.
  3. Figure 1: It is not clear what the fitted line indicated by “log” represents. Is the purpose to model accuracy (y) as a function of the logarithm of area? If that is the case, consider transforming the x-axis to the logarithm of area, and perhaps this will provide a simple linear relationship that can be modeled easily by a straight line.
  4. Table 6. Are units of acres acceptable by the journal? Convert to square kilometers?
  5. L607: Adjusting a map to spatially re-allocate a class based on the area estimated from a reference sample was proposed by Song et al. (2017) so there is previous support for the approach the authors suggest.

Song, X.-P., Potapov, P. V., Krylov, A., King, L., Di Bella, C. M., Hudson, A., Khan, A., Adusei, B., Stehman, S. V., and Hansen, M. C. (2017). National-scale soybean mapping and area estimation in the United States using medium resolution satellite imagery and field survey.  Remote Sensing of Environment, 190, 383-395.

  1. L654-655: It is my understanding that NLCD has reference sample data to assess accuracy of the NLCD land cover products. Has there been a comparison of CDL to NLCD reference data?
  2. L669: replace select with “selects”
  3. L722: extra “more” in “corn is more mapped more confidently”

Reviewer 3 Report

Line 100 please include Resourcesat-2 LISS and Sentinel 2 a/b

Line 258 - 2012 was a challenging year, as only DMC data was available for imaging. Also challenging was the large drought that impacted ag production across the Midwest and not much was mentioned in the paper about that. 

320 not sure if its possible to list/label some outlier states on figure?

341 the 2012 CDL used the original 2006 NLCD. Did this analysis use the 2006 original or reprocessed?

355 "were confusion" grammar

454 table 6, harvested cotton was published in 2012 August Production Report.

461 FSA reference data? Do you mean CDL mapping bias?

462 the way this sentence is worded, it sounds like the bias is worsening, in fact this shows it rose to .4 and .5, not bad when compared to other crops.

504 USDA data sources such as?

572 forage crop

***************

This research focused on year 2012. Perhaps 2017 would also have served as a better benchmark year, for a few reasons. It was a very good ag production year, the same census, survey, and NRI data were available for assessment and most importantly there were multiple medium resolution satellite platforms available in that year. 2012 was a catastrophic agriculture production year not only because of drought but there was very limited medium resolution satellites for ag monitoring. Additionally, as noted by the author, the CDL accuracy has improved over time.

There have been many industry discussions and ongoing research on quantifying land cover/land use change regarding ag expansion into sensitive ecosystems. The CDL provides an annual snapshot of production agriculture and questions also get asked about assessing grasslands and the non-ag domain. This paper demonstrated multiple formula and metrics to assess certainty across time and space for measuring LC/LU change in both the ag and non-ag domains.

This paper did a good job explaining methods and equations used to aggregate and consolidate classes. The paper explained the purpose and method for each equation and evolved into more complex equations with adequate explanation and discussions. Class consolidation supported the notion that the smaller acreage crops have lower accuracies while the large area crops perform better. It was novel to see the accompanying confidence layer brought into the discussion to get at spatial variation in CDL performance. It proved the assumption that class accuracies degrade as one moves away from the Heartland region of the country. As you move away from this area, the fields tend to get smaller, crops are more diverse, and farmer reporting/participation drops.

Figure 3 showed an interesting progression on accuracy and uncertainty that are useful for mapping assessments.

The aggregation process served the purpose of measuring LC/LU change, as it reduced emphasis on spectrally similar classes and detected change when there is potential land conversion, thereby providing data users a means to assess accuracy on change.

The limitations and uncertainty discussion was accurate and methods/metrics were appropriate.

The CDL scale and aggregation process provided the ability to separate crop/non crop across space/time. A unique approach was provided with bias correction and usage of the confidence layer, thereby giving much needed assessment techniques for the ag industry to identify/quantify LC/LU changes and land conversions.

Nicely written and very informative!

Round 2

Reviewer 1 Report

I am satisfied with the authors' responses.